# Personalized computational heart models with T1-mapped fibrotic remodeling predict sudden death risk in patients with hypertrophic cardiomyopathy

Ryan P O'Hara[1], Edem Binka[2], Adityo Prakosa[1], Stefan L Zimmerman[3], Mark J Cartoski[4], M Roselle Abraham[5], Dai-Yin Lu[5], Patrick M Boyle[6], Natalia A Trayanova[1]*

[1]Department of Biomedical Engineering, Johns Hopkins University, Baltimore, United States; [2]Division of Pediatric Cardiology, Johns Hopkins School of Medicine, Baltimore, United States; [3]Department of Radiology, Johns Hopkins School of Medicine, Baltimore, United States; [4]Division of Pediatric Cardiology, Nemours/Alfred I. duPont Hospital for Children, Wilmington, United States; [5]Division of Cardiology, University of California, San Francisco, San Francisco, United States; [6]Department of Bioengineering, University of Washington, Seattle, United States

**Abstract** Hypertrophic cardiomyopathy (HCM) is associated with risk of sudden cardiac death (SCD) due to ventricular arrhythmias (VAs) arising from the proliferation of fibrosis in the heart. Current clinical risk stratification criteria inadequately identify at-risk patients in need of primary prevention of VA. Here, we use mechanistic computational modeling of the heart to analyze how HCM-specific remodeling promotes arrhythmogenesis and to develop a personalized strategy to forecast risk of VAs in these patients. We combine contrast-enhanced cardiac magnetic resonance imaging and T1 mapping data to construct digital replicas of HCM patient hearts that represent the patient-specific distribution of focal and diffuse fibrosis and evaluate the substrate propensity to VA. Our analysis indicates that the presence of diffuse fibrosis, which is rarely assessed in these patients, increases arrhythmogenic propensity. In forecasting future VA events in HCM patients, the imaging-based computational heart approach achieved 84.6%, 76.9%, and 80.1% sensitivity, specificity, and accuracy, respectively, and significantly outperformed current clinical risk predictors. This novel VA risk assessment may have the potential to prevent SCD and help deploy primary prevention appropriately in HCM patients.

**\*For correspondence:**
ntrayanova@jhu.edu

**Competing interest:** The authors declare that no competing interests exist.

## Introduction

Hypertrophic cardiomyopathy (HCM) is the most common cause of sudden cardiac death (SCD) in the young and is a significant cause of sudden death in adults (*Maron, 2004*). The disease, with an incidence of 1 in 500, presents with progressive myocardial fibrosis which can create substrates for ventricular arrhythmias (VAs) leading to SCD in patients who are typically asymptomatic (*Galati et al., 2016*; *Olivotto et al., 2012*). Implantable cardioverter defibrillator (ICD) deployment, a procedure that carries risk of potential complications and morbidity, is used as primary prevention of SCD due to VA in patients with HCM (*Lambiase et al., 2016*; *Jayatilleke et al., 2004*). However, current risk stratification criteria outlined by the American College of Cardiology Foundation (ACCF)/American Heart Association (AHA) and European Society of Cardiology (ESC) fail to accurately identify all patients at risk for SCD, leading to suboptimal rates of appropriate ICD implantation (*Gersh et al.,*

*2011*; *O'Mahony et al., 2014*; *Schinkel et al., 2012*). Thus, many HCM patients receive ICDs without deriving any health benefits, while others are not adequately protected. The development of accurate means to stratify SCD risk due to VA in HCM patients for guidance of ICD deployment is an important unmet clinical need.

Cardiac magnetic resonance (CMR) imaging with late gadolinium enhancement (LGE) has unparalleled capability in the detection and quantification of scar and dense fibrosis (*Prakosa et al., 2014*). In HCM, myocardial fibrosis takes the form of both dense (focal) and diffuse fibrosis, with histopathological evidence showing diffuse fibrosis as the hallmark feature of the disease (*Galati et al., 2016*). Diffuse fibrosis, however, is not well captured by standard LGE-CMR. Instead, postcontrast T1 mapping, a parametric imaging modality, has been used to visualize diffuse fibrosis in patients with HCM (*Chu et al., 2017*; *Ellims et al., 2012*). We have previously developed a computational modeling approach (virtual heart) to predict SCD risk due to VA in postinfarction patients (*Arevalo et al., 2016*). We hypothesized that a new personalized virtual-heart technology, one that entails constructing fusion electrophysiological models based on the distribution of both dense and diffuse fibrosis, as acquired by the two different CMR modalities, would be predictive of the propensity of the HCM-remodeled substrate to VAs and could thus be used to assess SCD risk due to VA in this patient population.

The goal of this study is to create a personalized virtual-heart approach based on the combination of postcontrast T1 mapping and LGE-CMR and to employ it (1) to analyze how HCM-specific remodeling promotes arrhythmogenesis and (2) in a targeted strategy to forecast risk of VA in HCM patients. In a proof-of-concept patient cohort, we assess the predictive capability of the approach as compared to that of other clinical metrics for VA risk prediction in HCM.

## Results

The new approach to analyzing arrhythmogenic propensity in HCM patients developed here involved creating three-dimensional (3D) patient-specific electrophysiological ventricular models based on fusing data from LGE-CMR and postcontrast T1 mapping. Each model thus represented the personalized distribution of focal fibrosis (scar) and diffuse fibrosis. VA inducibility in each HCM patient's substrate was probed to determine VA risk for the patient and to understand the mechanisms of arrhythmogenesis, and specifically, the contribution of the individualized diffuse fibrosis distribution, which is rarely assessed in these patients. Conceptual overview of our approach to analyzing the arrhythmogenic propensity of HCM patient hearts is presented in *Figure 1A*.

### Patient characteristics

Twenty-six patients with HCM were included in this study. Demographic information for the cohort is provided in *Table 1*. All patients were adults (median age 53 years) and our cohort was 19% female. Thirteen of the 26 HCM patients experienced clinical VAs. Of the clinical parameters that associate with SCD in HCM (FHSCD, unexplained syncope, MWT, Max LVOTG, age, and LA diameter; see *Table 1* for abbreviations), there were no statistically significant differences (p = 0.34, –, 0.65, 0.72, 0.98, 0.26) between patients with and without clinical VA. There was no statistically significant difference in any of the other common clinical characteristics between the two groups. Clinical data alone were not sufficient to accurately determine VA risk in this population.

Values are given as *n* (%), mean [range], or mean ± standard deviation (SD). p values were calculated using Student's *t*-test (p ≤ 0.05 considered statistically significant). VA = ventricular arrhythmia; CMR = cardiac magnetic resonance; NYHA = New York Heart Association; ASA = alcohol septal ablation; AF = atrial fibrillation; LA = left atrium; LVOTG = left ventricular outflow tract gradient; MWT = maximum wall thickness; FS = fractional shortening; FHSCD = family history of sudden cardiac death.

### Assessment of HCM structural remodeling using LGE-T1 geometrical models

To reconstruct the geometrical model of each patient's heart, LGE-CMR and postcontrast T1 mapping images were combined, creating a personalized LGE-T1 fusion model of HCM ventricular geometry and structural remodeling. *Figure 1B* presents the 'fusing' process, in which an initial reconstruction of ventricular geometry and scar/fibrosis was performed from the LGE-CMR images using standard 'one-size-fits-all' thresholds, and then the relaxation times from the short-axis T1 map were used

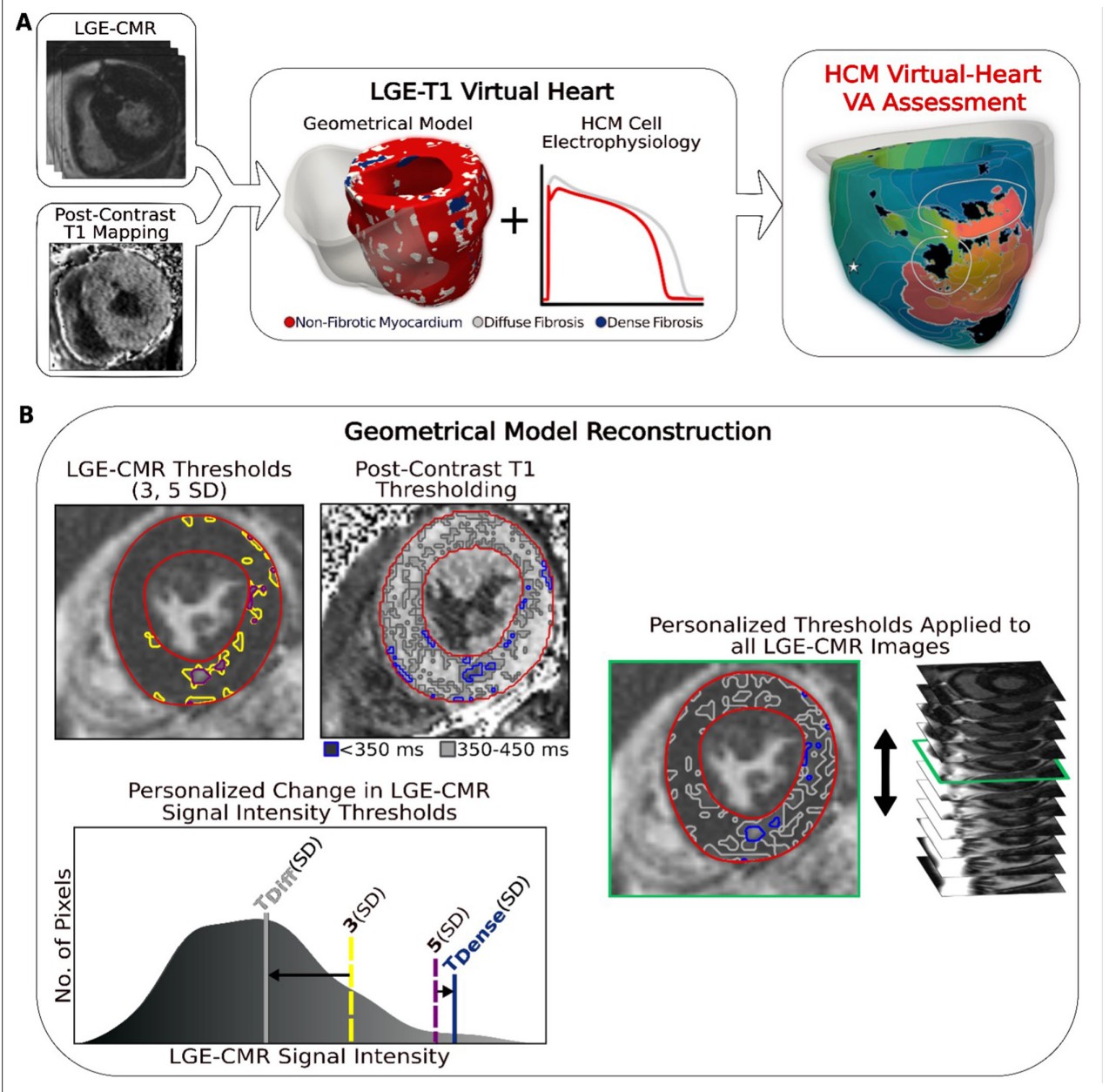

**Figure 1.** Overview of the approach and geometrical model reconstruction. (**A**) Flowchart summarizing the virtual-heart ventricular arrhythmia (VA) risk stratification approach for hypertrophic cardiomyopathy (HCM) patients. A combination of late gadolinium enhancement (LGE)-cardiac magnetic resonance (CMR) and postcontrast T1 mapping is used to construct personalized left ventricular (LV) geometrical models with fibrotic remodeling. Incorporating HCM-specific electrophysiological properties (action potential kinetics, conduction velocity) completes the generation of each personalized LGE-T1 virtual heart, which is then used to assess VA propensity in the substrate via rapid pacing. right ventricle (RV) is shown in transparent gray. Dense fibrosis (scar) is considered nonconductive. (**B**) Fusing LGE-CMR and postcontrast T1 map information to generate the personalized, geometrical virtual-heart model. Top left: LV segmentation with intermediate and high signal intensity thresholds of 3 (yellow) and 5 standard deviation (SD) (purple), respectively, on short-axis LGE-CMR. Bottom left: mid-ventricular postcontrast T1 map segmentation with relaxation time thresholds of <350 (blue) and 350–450 ms (gray). Right: the thresholds of the LGE-CMR signal intensity were adjusted to new, personalized thresholds, $T_{Diffuse}$ and $T_{Dense}$, based on the T1 map (see text for detail). The new personalized signal intensity thresholds in the matching LGE-CMR slice were then applied to all LGE-CMR short-axis slices.

**Table 1.** Patient characteristics (*N* = 26).

| Clinical characteristic | Patients without VA, *n* = 13 | Patients with VA, *n* = 13 | p value |
|---|---|---|---|
| Male | 12 (92) | 9 (69) | 0.08 |
| Age at CMR, years | 49.7 [19–76] | 49.8 [22–78] | 0.98 |
| NYHA III/IV | 4 (31) | 4 (31) | – |
| Myectomy | 1 (8) | 1 (8) | – |
| ASA | 1 (8) | 2 (15) | 0.34 |
| Amiodarone | 0 (0) | 1 (8) | 0.34 |
| Persistent AF | 3 (23) | 4 (31) | 0.34 |
| LA diameter, mm | 43.8 ± 6.3 | 38.3 ± 12.7 | 0.26 |
| Max LVOTG, mm Hg | 57.8 [4–154] | 50.8 [8–160] | 0.72 |
| MWT, mm | 20.5 ± 5.0 | 19.6 ± 5.6 | 0.65 |
| FS, % | 38.0 ± 10.2 | 40.3 ± 10.8 | 0.40 |
| FHSCD | 3 (23) | 4 (31) | 0.34 |
| Unexplained syncope | 3 (23) | 3 (23) | – |

to define *personalized* signal intensity thresholds to delineate areas of diffuse fibrosis and scar (see Methods for detailed description). The personalized thresholds were unique to each patient. The additional personalization of the geometrical model furnished by the usage of the T1 mapping data ensured a comprehensive representation of the individualized structural remodeling in each patient's heart.

Once the geometrical models were reconstructed, they were analyzed to determine whether the level (amount) and/or distribution of structural remodeling discriminate between patients with and without clinical VA. The level of regional hypertrophy was first assessed, as measured by the wall thickness of the heart models. No statistically significant difference in regional hypertrophy was found at the septum (p = 0.61), anterior wall (p = 0.84), posterior wall (p = 0.94), and apex (p = 0.73) between heart models of patients with and without clinical VA, as shown in *Table 2*. These results indicated that the level of hypertrophy does not discriminate between arrhythmogenic and nonarrhythmogenic substrates in HCM patients.

*Figure 2A and B* presents a comparison between geometrical heart models of two patients (one with clinical VA and another without) reconstructed by combining LGE-CMR with T1 mapping, and by using LGE-CMR only. In the latter models, the accepted 'one-size-fits-all' thresholds of three and five times the SD of the low-intensity mean were used to reconstruct dense fibrosis (scar) and diffuse fibrosis distributions (see Methods). In the former models, patient-specific thresholds from the T1 mapping were used to delineate dense and diffuse fibrosis. As evident from the figure, using patient-specific signal intensity thresholding from the T1 map resulted in a significantly higher amount of diffuse fibrosis in these two models (42.9 ± 3.4% vs 9.8 ± 0.1%).

For all HCM LGE-T1 fusion models, the average threshold for diffuse fibrosis, $T_{Diffuse}$, was 1.1 ± 0.7, significantly different from the corresponding LGE 'one-size-fits-all value', 3 SD. The average threshold for dense fibrosis, $T_{Dense}$, was 5.1 ± 0.5, not a significant change from the original 5 SD. The personalized threshold adjustment did not therefore result in a significant change in the amount of dense fibrosis for LGE-T1 models compared to LGE-only models (averages of 3.8 ± 2.3 vs 3.2 ± 1.3, p = 0.30). However, it resulted in a significant change in diffuse fibrosis across all models, as illustrated in *Figure 2C* (40.5 ± 9.4% for LGE-T1 vs 8.9 ± 1.7% for LGE only, p < 0.0001).

No statistical differences were found in the amounts of diffuse fibrosis between LGE-T1 models with and without clinical VA (p = 0.53, confidence interval [CI: 36.8, 44]) and between LGE-only models with and without clinical VA (p = 0.94, CI: [8.25, 9.53]; *Figure 1B*); also, no statistical difference was found in the amount of scar (3.7 ± 2.2% vs 3.8 ± 2.5%, p = 0.91 for LGE-T1 and 3.4 ± 1.2% vs 3.0 ± 1.5%, p = 0.53 for LGE-only models). These results indicate that the imaging characteristics of HCM structural remodeling, as

**Table 2.** Left ventricular (LV) wall thickness in hypertrophic cardiomyopathy (HCM) patients with and without clinical ventricular arrhythmia (VA).

| | Patients without VA, *n* = 13 | Patients with VA, *n* = 13 | p value |
|---|---|---|---|
| Wall thickness (mean ± standard deviation [SD]) | | | |
| Septum, mm | 11.3 ± 8.4 | 13.1 ± 9.2 | 0.61 |
| Anterior, mm | 11.3 ± 7.4 | 10.7 ± 7.3 | 0.84 |
| Posterior, mm | 11.1 ± 7.2 | 11.3 ± 7.3 | 0.94 |
| Apex, mm | 7.9 ± 5.3 | 7.2 ± 4.8 | 0.73 |

p values were calculated using Student's *t*-test (p ≤ 0.05 considered statistically significant).

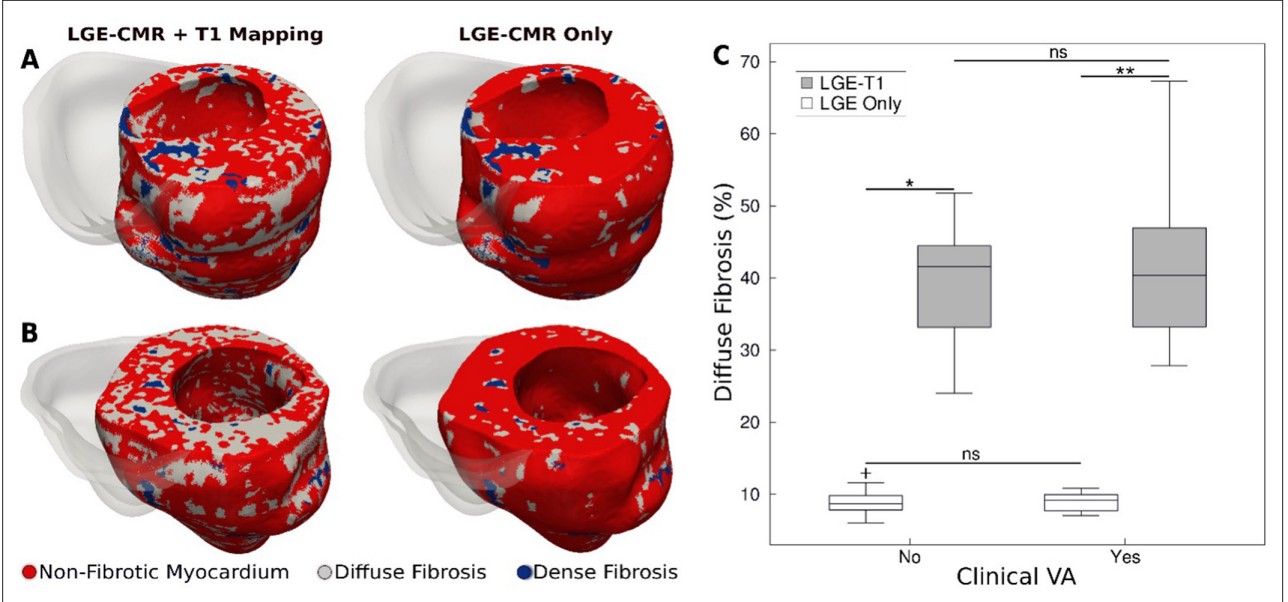

**Figure 2.** Fibrotic remodeling. (**A,B**) Examples of hypertrophic cardiomyopathy (HCM) personalized left ventricular (LV) geometrical models with fibrotic remodeling (right ventricle [RV] shown in transparent gray) reconstructed using late gadolinium enhancement (LGE)-cardiac magnetic resonance (CMR) images with personalized T1-informed fibrosis segmentation thresholds (left) and using LGE-CMR images with one-size-fits-all fibrosis segmentation thresholds of 3 and 5 standard deviation [SD] (right). There is significantly more diffuse fibrosis in the T1-adjusted models. (**A**) Heart model from an HCM patient without clinical ventricular arrhythmia (VA). (**B**) Heart model from an HCM patient with clinical VA. (**C**) Boxplot of the amount of diffuse fibrosis in LGE-T1 and LGE-only HCM geometrical models without clinical VA (LGE-T1: *N* = 13, interquartile range [IQR] = 12.54; LGE only: *N* = 13, IQR = 2.41; *p < 0.0001) and with clinical VA (LGE-T1: *N* = 13, IQR = 14.44; LGE only: *N* = 13, IQR = 2.46; **p < 0.0001). The '+' denotes an outlier.

The online version of this article includes the following source data for figure 2:

**Source data 1.** Spreadsheet including source data underlying *Figure 2*.

visualized by the combination of LGE-CMR and T1 mapping, cannot be used to discriminate between patients who will and will not develop clinical VA.

For each geometrical model used in this study, the amount of diffuse fibrosis in each LGE-T1 and LGE-only model (*Figure 2—source data 1*).

## Assessment of propensity to VA in HCM LGE-T1 virtual-heart models

Once the geometrical models of all HCM patients were reconstructed, electrophysiological models were generated and used to assess the individualized propensity to VA by pacing from distributed ventricular sites, representing potential ectopy. Full detail is in Methods. A total of 182 simulations ([26 patient heart] × [7 pacing locations]) were performed to probe propensity to VA induction in this cohort. To be able to better understand the role of T1-based diffuse fibrosis in arrhythmogenesis, we also repeated the simulations with LGE-only models.

*Figure 3* presents reentrant arrhythmias induced (from sites marked with stars) in three LGE-T1 virtual hearts from patients with known clinical VAs. In all three cases, a single VA morphology was induced. In *Figure 3*, left, the VA localized in a region of interdigitated diffuse and dense fibrosis. In *Figure 3*, middle, there was a figure-of-eight reentry in a transmural region of diffuse fibrosis. In *Figure 3*, right, the VA shown was induced from two different pacing sites, one in the basal lateral and another in the inferoseptal wall and persisted also in a region of interdigitated diffuse and dense fibrosis.

For each geometrical model that reentry was induced, the number of unique VA morphologies and amount of diffuse fibrosis in each LGE-T1 and LGE-only model (*Figure 4—source data 1*).

Of the 26 LGE-T1 models, 14 were found inducible for VA in simulations. In contrast, only 12 LGE-only models were found inducible, indicating that the presence of diffuse fibrosis leads to increased VA inducibility. *Figures 4 and 5* explore the mechanistic contributions to increased VA vulnerability in models with T1-based diffuse fibrosis.

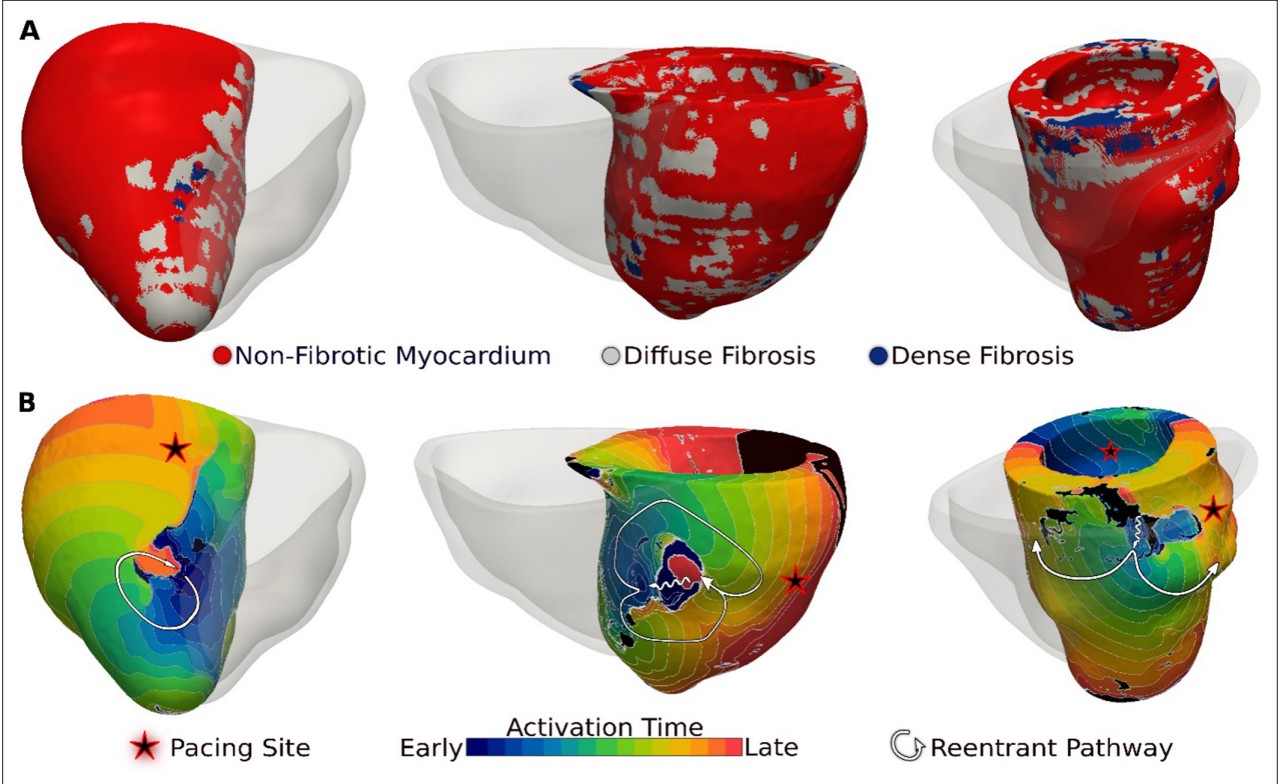

**Figure 3.** Ventricular arrhythmias (VAs) induced in three late gadolinium enhancement (LGE)-T1 virtual hearts from patients with known clinical VAs. (**A**) Reconstructed patient-specific geometrical models. (**B**) Activation patterns of the reentry induced from the pacing site(s) marked with star. Black regions did not activate during the timeframe shown.

*Figure 4A* presents the number of unique VA morphologies induced by the pacing protocol. LGE-T1 models had a total of 24 unique VAs induced in them (out of total 32 VA episodes induced in the LGE-T1 models); in each model, there were between 1 and 3 different VA morphologies. LGE-only models had a total of 15 unique VAs (with a total of 17 VA episodes induced in these models), with only 1 or 2 distinct VA morphologies induced per model. These results indicate that the presence of diffuse fibrosis as reconstructed from T1 mapping increases the number of unique VAs in each substrate. *Figure 4B* correlates the amount of diffuse fibrosis and the number of unique VAs ($R = 0.40$, p = 0.157) in LGE-T1 inducible models. The moderate negative correlation indicates that the distribution of diffuse fibrosis is more important than its amount as the mechanism of VA inducibility in the HCM-remodeled substrate. *Figure 4C* presents two bullseye plots with the 7 AHA regions in which pacing sites were located; shown are the number of pacing sites in each segment that elicited VAs in LGE-T1 and LGE-only inducible models. In the LGE-T1 models, out of the 98 pacing sites (7 pacing sites per each of the 14 inducible models), 32 (33%) resulted in VA induction. In contrast, out of 84 pacing sites in the 12 LGE-only inducible models, 17 (20%) resulted in VA induction. Thus, the presence of T1-based diffuse fibrosis renders the substrate inducible from a larger number of ectopic locations, contributing to the overall increased vulnerability to VA. Interestingly, the sector with the pacing sites that induced most VAs (mid-anteroseptal) and that with least (basal inferolateral) were the same in LGE-T1 and LGE-only models, indicating that the additional T1-based diffuse fibrosis localizes to the sectors with arrhythmogenic substrate in the LGE-only models. Overall, the distribution of pacing sites is the same (with small exception in the basal regions), but the number of sites per sector increased with the presence of diffuse fibrosis.

*Figure 5* explores the contribution of T1-based diffuse fibrosis to VA inducibility by comparing arrhythmogenesis in individual models. Panel A shows, for the 13 patients with clinical VAs, the number of distinct VAs per patient model. First, LGE-T1 modeling documented correctly VA occurrence in the digital substrates of 11 of the 13 patients with clinical VAs (compared to 9 LGE-only correct VA predictions); thus, the representation of T1-based diffuse fibroses increased the fidelity of

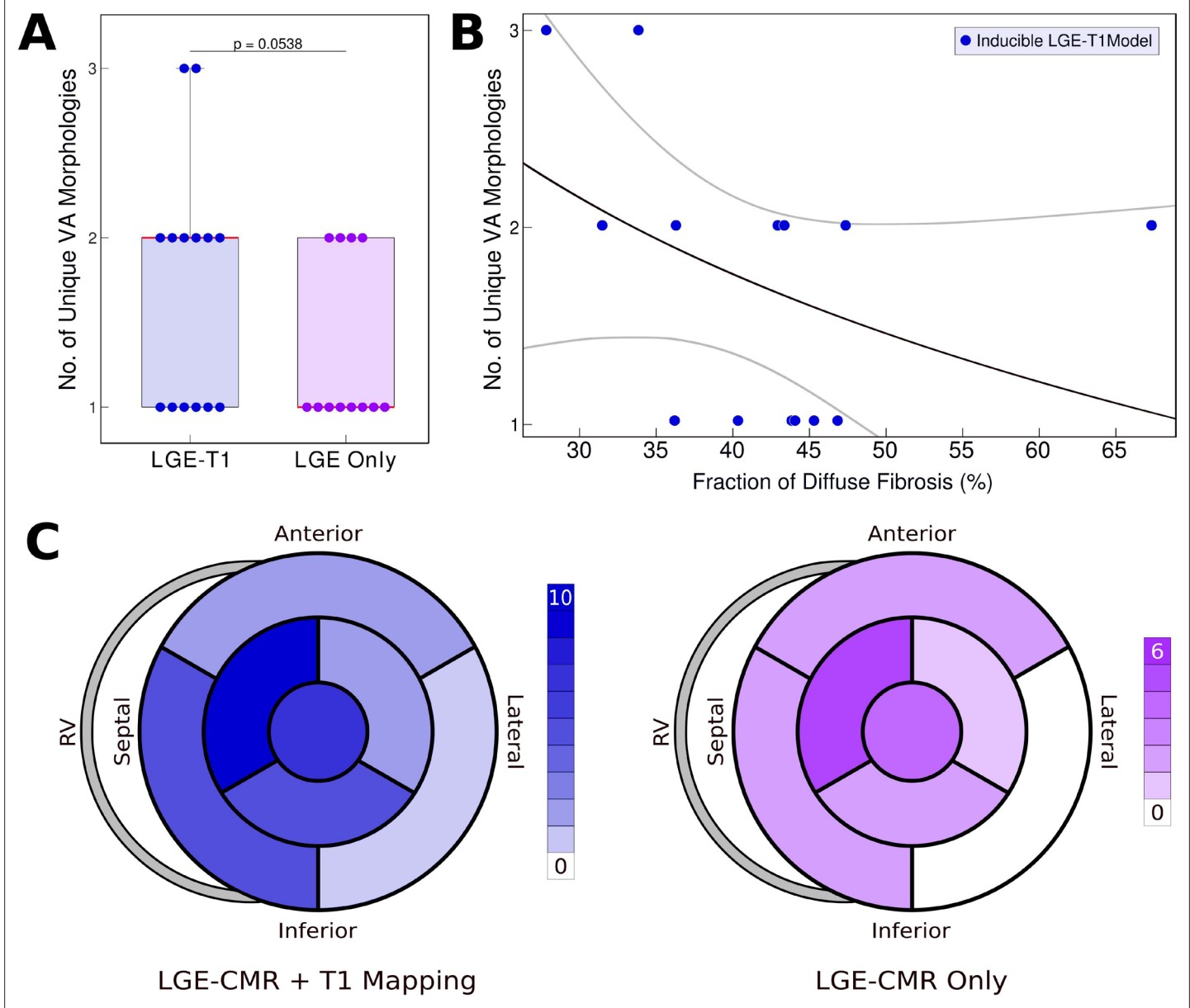

**Figure 4.** Relationship between T1-based diffuse fibrosis and ventricular arrhythmia (VA) inducibility in late gadolinium enhancement (LGE)-T1 and LGE-only personalized virtual-heart models of hypertrophic cardiomyopathy (HCM) patients. (**A**) Comparison of the number of unique VA morphologies between inducible LGE-T1 and LGE-only models for all VA-inducing pacing sites (LGE-T1: N = 14, interquartile range [IQR] = 1; LGE only: N = 12, IQR = 0.75; p = 0.0538, confidence interval [CI]: [1.25, 1.75]). (**B**) Correlation between amount of T1-based diffuse fibrosis and the number of unique VA morphologies induced in LGE-T1 models using logistic regression (R = 0.40, p = 0.157). (**C**) The distribution of the pacing sites that induced VAs in LGE-T1 and LGE-only models.

The online version of this article includes the following source data for figure 4:

**Source data 1.** Spreadsheet including source data underlying *Figure 4*.

the HCM virtual-heart approach. Second, while the number of unique VA morphologies per inducible virtual heart increased in LGE-T1 models as compared to LGE only (consistent with data in *Figure 4A*), with LGE-T1 models having maximum three unique VAs (vs 2 in LGE only) in the cohort, this plot points to interesting VA dynamics in individual substrates. Patients 5, 6, and 9 had the same level of arrhythmogenicity of the substrate (1 VA induced) regardless of the presence of T1-based diffuse fibrosis. An example of VA dynamics in these models is shown in *Figure 5B* (patient 6). The additional diffuse fibrosis did not alter the location or direction of VA reentry; the reentry occurred in a region of dense

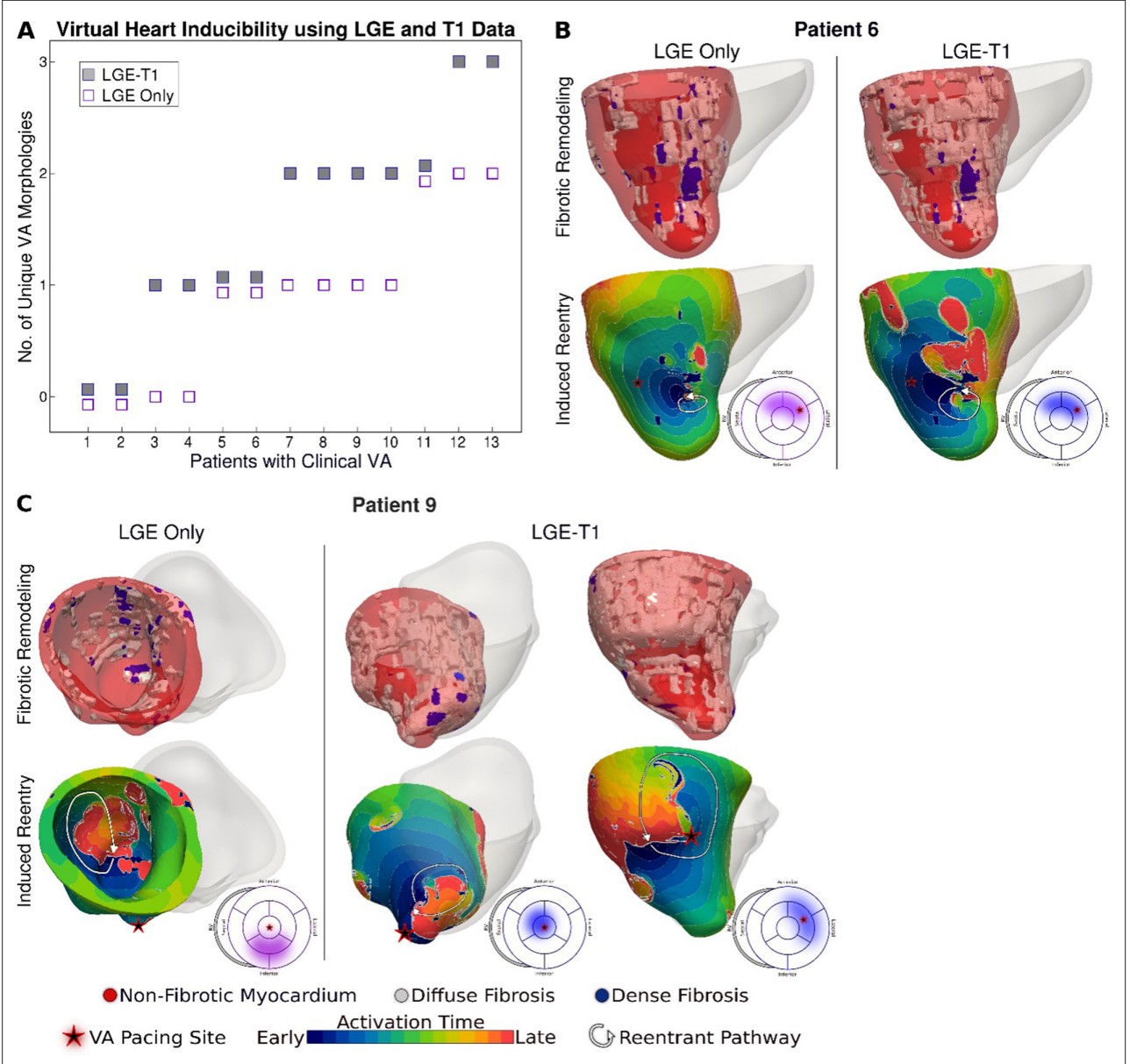

**Figure 5.** Comparison of arrhythmogenesis in hypertrophic cardiomyopathy (HCM) models of patients with clinical ventricular arrhythmias (VAs). (**A**) Plot of the number of unique VA morphologies for patients with clinical VA using late gadolinium enhancement (LGE)-T1 and LGE-only models. (**B, C**) Comparison of VAs in corresponding LGE-T1 and LGE-only models. Pacing site(s) are marked with stars. Bullseye plots show the pacing site location (star) and the location of the reentrant pathway in LGE-T1 (blue) and LGE-only (purple) models.

The online version of this article includes the following source data for figure 5:

**Source data 1.** Spreadsheet including source data underlying *Figure 5*.

scar in both types of models. Although the T1-based diffuse fibrosis (36.2%) in this patient augments the existing diffuse fibrosis of the LGE-only model (9.6%), there are no additional VA morphologies. However, the activation dynamics were altered, with propagation being less organized.

In the reminder of the clinical-VA patients in *Figure 5A*, the presence of T1-based diffuse fibrosis resulted in the occurrence of additional VA(s), on top of that (those) also present in the LGE-only model. Furthermore, there was not a strict correspondence between the VAs in the LGE-only models and the equivalent ones in the LGE-T1 virtual heart in terms of VA locations and dynamics. This indicates that the arrhythmogenic substrate changes in a global fashion when T1-based diffuse fibrosis in considered. An example is presented in *Figure 5C* (patient 9).

**Table 3.** Predictive capability of hypertrophic cardiomyopathy (HCM) virtual-heart technology.

|  | Sensitivity | Specificity | PPV | NPV | Accuracy |
|---|---|---|---|---|---|
| ACCF/AHA risk model | 46.2 | 46.2 | 46.2 | 46.2 | 46.2 |
| ESC risk model | 53.9 | 38.5 | 46.7 | 45.5 | 46.2 |
| Virtual-heart technology: LGE-T1 | 84.6 | 76.9 | 78.8 | 83.3 | 80.1 |
| Virtual-heart technology: LGE only | 69.2 | 76.9 | 75.0 | 71.4 | 73.1 |

ACCF = American College of Cardiology Foundation, AHA = American Heart Association, ESC = European Society of Cardiology, LGE = late gadolinium enhancement.

As VAs in HCM patients can occur under different circumstances and be documented by different means, there is no invasive clinical mapping data for these patients regarding the location and morphologies of the clinical VAs. However, two patients in this cohort underwent clinical electrophysiology studies that identified episodes of VA. The simulated VAs in the LGE-T1 models matched the locations of the clinical VAs as documented by the chart review.

It remains unclear why patients 1 and 2 had noninducible LGE-T1 substrates despite having clinical VAs. The amount of diffuse fibrosis in these two substrates (42.4 ± 12.2%) was similar to that in the 11 inducible substrates (41.2 ± 10.9%); similar was the finding regarding dense scar. Additionally, the maximum wall thickness and the thickness distribution fall into the same ranges as the averages of the cohort. It is possible that there have been other factors, including electrophysiological remodeling that the fusion substrate-based modeling approach presented here cannot capture. Despite incorrect prediction in 2 out of 26 patients in the HCM cohort, the VA risk prediction capabilities of the LGE-T1 virtual-heart approach significantly surpassed those of any current clinical risk assessment approaches, as detailed in the next section.

For each geometrical model with clinical VA, the number of unique VA morphologies in each LGE-T1 and LGE-only model (*Figure 5—source data 1*).

## Assessment of the capability of the HCM virtual-heart VA risk prediction

After examining the mechanistic underpinning of arrhythmogenesis and the role of T1-based diffuse fibrosis in the HCM substrate, we conducted a comparison of our VA risk predictor capability with the clinical risk assessment guidelines of the ACCF/AHA and ESC. Results are presented in *Table 3*, illustrating that both existing clinical approaches were significantly inferior in predicting VA risk in this cohort. Of the 13 HCM patients with clinical VAs, the ACCF/AHA model predicted correctly 6 of the patients, while the ESC model predicted correctly 7 patients; the LGE-T1 virtual-heart approach predicted correctly 11 patients. Overall, our LGE-T1 virtual-heart technology exhibited higher accuracy and greater sensitivity and specificity (80.1%, 84.6%, and 76.9%) as compared to the best performing corresponding metrics of the clinical risk assessment methodologies (46.2%, 53.9%, and 46.2% for accuracy, sensitivity, and specificity, respectively).

For completeness, data at the bottom of *Table 3* quantify the predictive capability of the substrate arrhythmogenesis approach when using LGE-only models (9 patients predicted correctly out of 13). Interestingly, even without the additional T1 personalization (i.e. without accounting for T1-based diffuse fibrosis), the LGE-only virtual-heart technology outperformed the clinical risk stratifiers in this HCM cohort. This finding indicates that assessing the arrhythmogenic propensity of the substrate is of paramount importance to HCM VA risk stratification, even when the distribution of diffuse fibrosis may not be accurately represented.

## Discussion

In this study, we presented a new personalized virtual-heart approach for assessing arrhythmia risk in patients with HCM, which could be used in guiding clinical decisions for prophylactic ICD implantation. Our technology uses multiscale computational models of patients' hearts reconstructed on the basic of the fusion of imaging data from LGE-CMR and T1 mapping. With the inclusion of

information from postcontrast T1 mapping, a quantitative and parametric imaging modality, extensive diffuse fibrotic remodeling, which is a hallmark of HCM, is adequately represented. Here, this is done by adjusting the diffuse fibrosis intensity-based thresholds in model construction based on the T1 maps, while preserving the identification of dense scar from LGE-MRI. These are the first personalized heart models created with data from different types of CMR; previous personalized modeling approaches for VA assessment have utilized LGE-MRI scans in reconstructing model geometry/ structure (*Prakosa et al., 2014*; *Cartoski et al., 2019*; *Arevalo et al., 2016*). Once constructed, the mechanistic personalized electrophysiological models were used to analyze how HCM-induced remodeling, and specifically the presence of diffuse fibrosis promotes arrhythmogenesis. Finally, the capability of our approach to forecast future VA events was assessed in the cohort of 26 HCM patients.

The presence of diffuse fibrosis has been suggested previously as a potential factor in increased risk of VA, in addition to focal scar. Previous studies have found associations between diffuse fibrosis and VA in nonischemic dilated cardiomyopathies (*Nakamori et al., 2018*) and mitral valve prolapse (*Bui et al., 2017*). Additionally, diffuse ventricular fibrosis has been found to increase left atrial pressure, and may be a marker of atrial fibrillation recurrence postablation (*Begg et al., 2020*). However, its contribution to arrhythmogenic propensity in HCM patients has never been assessed before. This study found that the presence of T1-based diffuse fibrosis resulted in the occurrence of new VAs, in addition to those arising from scar (as assessed by signal heterogeneities in LGE-MRI). Furthermore, T1-based diffuse fibrosis distribution rendered the substrate inducible from a larger number of ectopic locations, contributing to the overall increased vulnerability to VA. The arrhythmogenic propensity of the ventricular substrate with diffuse fibrosis is patient specific; it is the amount of fibrosis and its distribution (shape and location) that determine inducibility of arrhythmia.

In this retrospective proof-of-concept HCM study, the personalized LGE-T1 virtual-heart technology demonstrated excellent performance in forecasting future VA events in HCM patients, achieving 84.6%, 76.9%, and 80.1% sensitivity, specificity, and accuracy, respectively. It outperformed both risk models used in current clinical practice, the ACCF/AHA and ESC models. Indeed, all 26 HCM patients in our study were deemed at high risk for SCD by the ACCF/AHA criteria and received ICDs for primary prevention, but only 13 patients, that is 50% of the cohort, actually experienced VA (appropriate ICD firing). Should our LGE-T1 virtual-heart technology be proven to be a superior risk predictor in larger clinical studies, it would advance the management of patients with this complex disease, helping to ensure that those at high risk for VA are adequately protected by ICDs and that unnecessary ICD implantations and the associated device complications are minimized.

The HCM virtual-heart technology's ability to comprehensively evaluate substrate arrhythmogenicity, as probed by rapid pacing delivered at a number of uniformly distributed ventricular locations, is paramount to its superior performance. Even when using only LGE-CMR in model construction, which reliably detects focal scar (dense fibrosis) but underestimates the amount of nonischemic fibrotic remodeling, our technology still offers VA risk assessment that is superior to the clinical risk models. However, the use of T1 maps in model construction confers a higher level of personalization in each patient's heart model as compared to LGE only (i.e. personalized thresholds for segmentation), which ultimately translates into superior predictive capability.

HCM is a genetic disease that progresses throughout the life of the patient, and a cardiac event might be a phenotypic expression of the disease at any point of time. Therefore, we envision that in the clinical application of our technology, patients would be reimaged at different time points and risk assessment repeated to account for changes in arrhythmia susceptibility over time as the diseased heart remodels.

The technology developed here charts a new direction in the use of biophysically detailed heart modeling in the prognosis of rhythm disorders. A number of different imaging modalities used in patient assessment such as positron emission tomography or single-photon emission computerized tomography could also be integrated with LGE-CMR to construct hybrid heart computational models. Combining such computational approaches with machine learning techniques (*Shade et al., 2020a*; *Shade et al., 2021*) will enable the incorporation of additional patient clinical data, such as genetic information, phenotypic characterization, time series, such as electrocardiography, and fibrotic distribution in the diagnosis and treatment of complex cardiac diseases with adequately sized patient cohorts.

## Study limitations

Our study has a small sample size, limited by the fact that a number of LGE-CMR scans of HCM patients had imaging artifact, which prevented us from reconstructing a larger number of virtual hearts. Specifically, aliasing and motion artifacts were main causes for excluding patient data as well as incomplete scans (operator did not scan the entire left ventricular [LV]). Further, there were minor discrepancies of the in-plane resolution between the postcontrast T1 map and the matching LGE-CMR short-axis scan which were mitigated by binning the regions of fibrotic remodeling and electrophysiological changes instead of using a continuum. The resolution of the CMR scan does not allow representation of small structural heterogeneities.

Finally, our virtual-heart technology does not incorporate cardiac mechanics into the arrhythmia risk stratification, however we do not expect that including it will alter our prediction. Indeed, a recent paper examining VA in a patient-specific model with structural remodeling that incorporated mechanics *Salvador et al., 2021* demonstrated that mechanics only modifies the stability of the arrhythmia, but does not alter its inducibility.

# Materials and methods

## Study overview

The methodology for assessing VA risk in HCM patients involves creating 3D patient-specific electrophysiological ventricular models with data from LGE-CMR and postcontrast T1 mapping. Each model represents the personalized distribution of both focal fibrosis (scar) and diffuse fibrosis, both of which contribute to the formation of the arrhythmogenic substrate. VA inducibility in each HCM patient's substrate is probed to understand the mechanisms of arrhythmogenesis, and specifically the contributions to it of the focal and diffuse fibrosis distributions, and to determine VA risk for the patient. Conceptual overview is presented in *Figure 1*.

The predictive capabilities of the virtual-heart HCM VA risk stratifier were evaluated retrospectively in a proof-of-concept study using data from 26 HCM patients. We chose a cohort that was balanced between patients with VAs based on appropriate ICD firings (13 patients) and without arrhythmic events (the other 13 patients). All patients underwent implantation of clinically indicated ICDs. Virtual-heart predictions of VA risk, executed blindly, were compared to clinical outcomes.

## Study population

The 26 patients were diagnosed with HCM based on the presence of LV wall thickness ≥15 mm on two-dimensional echocardiography in the absence of other ventricular diseases, including hypertrophy of the right ventricle (RV), between 2011 and 2016 at Johns Hopkins Hospital (*Chu et al., 2017*). All patients were clinically referred for prophylactic ICD implantation, being deemed at high risk for VA based on clinician assessment. T1 maps and LGE-CMR were obtained pre-ICD implantation. Patients were followed for the primary end point of appropriate ICD firing due to VA. As stated above, of the 26 HCM patients, 13 (50%) had known VA episodes based on appropriate ICD firing. Patient clinical characteristics are shown in *Table 1*.

## Imaging data

Patients whose imaging data were retrospectively used in this study had cardiac CMR examinations using a 1.5 T scanner (MAGNETOM Avanto; Siemens Healthcare, Erlangen, Germany) prior to ICD implantation. Short-axis LGE-CMR images with 2-mm in-plane and 8-mm *z*-axis resolution were acquired as previously described (*Chu et al., 2017*). In addition, a single mid-ventricular short-axis postcontrast T1 map with 1.5-mm in-plane and 8-mm *z*-axis resolution was acquired during the same scan 12 min after gadolinium injection using a MOLLI sequence (*Chu et al., 2017*). All patient imaging data for model generation were obtained under IRB approval.

## Geometrical reconstruction of patients' hearts from T1 maps and LGE-CMR images

In generating HCM patients' virtual hearts, a geometrical model of each patient's heart was first reconstructed using the patient's LGE-CMR images. The LV myocardium was segmented from short-axis LGE-CMR in the CardioViz3D software using a validated semiautomatic landmark-based method

to define the boundaries of the endocardium and epicardium as described in previous virtual-heart projects by our team (*Arevalo et al., 2016*; *Shade et al., 2020b*; *Cartoski et al., 2019*) The RV was not reconstructed due to blood pool artifacts and the lack of hypertrophy and fibrotic substrate in HCM patients. The LGE-CMR was processed in the standard manner for reconstructing LV geometrical models of patients with ischemic (*Arevalo et al., 2016*) or nonischemic cardiomyopathy (*Shade et al., 2020b*; *Cartoski et al., 2019*) which incorporate the distribution of scar and surrounding gray (border) zone. Specifically, as LGE-CMR is an image of relative intensity, the mean of the low signal intensity region in each image was determined, the latter representing myocardium without detectable fibrosis. The SD of that mean value was used to threshold regions of intermediate (>3 SD above the mean) and high (>5 SD above the mean) signal intensity in the LV, representing fibrotic gray zone and focal scar. *Figure 1B*, top left, shows these thresholds applied to one LGE-CMR image. The same SD thresholds were used for all models.

Information from the patient's postcontrast T1 mapping was next incorporated in each geometrical heart model. As only a single short-axis mid-LV postcontrast T1 map (*Figure 1B*) was acquired for each patient, the matching slice with the same position and orientation of the ventricle (*Figure 1B*, top left) in the LGE-CMR stack was selected using the z-axis coordinate of the images. Each set of a postcontrast T1 map and a corresponding short-axis LGE-MRI slice was visually inspected for differences in anatomy, cardiac phase, and distribution of enhancement, and only images found to be in agreement by the radiologists were used in this study. The postcontrast T1 map was segmented using the same method as described above. The relaxation times from the segmented LV of the short-axis T1 map were used to define new, *personalized* signal intensity thresholds (different from the 'one-size-fits-all' thresholds of 3 and 5 SD of the low-intensity mean) to delineate areas of intermediate and high signal intensities in the corresponding LGE slice. Specifically, the signal intensity profile of the myocardium from the corresponding LGE-CMR slice was normalized to the intensity profile (relaxation times) of the T1 map myocardium. Regions in the LGE-CMR slice signal intensity corresponding to short (<350 ms) and intermediate (350–450 ms) relaxation times in the T1 map were thresholded (*Figure 1B*). Based on evidence in histopathological studies (*Ellims et al., 2012*; *Iles et al., 2008*; *Mewton et al., 2011*; *Ellims et al., 2014*), these regions in the T1 map represent dense fibrosis (scar) and diffuse fibrosis.

The thresholds in the LGE-CMR slice, originally 3 and 5 SD of the mean signal intensity of the normal myocardium, were changed to new values ($T_{Diffuse}$ and $T_{Dense}$, in units of SD, *Figure 1B*, bottom left) such that the amount and distribution of tissue of mid- and high signal intensity in the LGE-CMR slice matched those in the T1 map. The new personalized signal intensity thresholds in the matching LGE-CMR slice were then applied to all LGE-CMR short-axis slices (*Figure 1B*, right) for the given patient to complete the generation of the LGE-T1 personalized geometrical heart model (*Figure 1A*, middle) that incorporates regions of focal scar and diffuse fibrosis; the personalized fibrosis segmentation thresholds were unique to each patient. Using T1 mapping provided additional personalization of the model geometrical reconstruction and ensured a comprehensive representation of the individualized structural remodeling in each patient's heart.

High-resolution finite-element tetrahedral meshes, with an average resolution of 355 ± 69 μm, were constructed directly from the ventricular segmented images using finite-element analysis software (Mimics Innovation Suite; Materialise, Leuven, Belgium); the software uses an input target finite-element edge length and generates a computational mesh with a tight edge length distribution around the input value. The mesh resolution requirement in electrophysiological simulations is 300–400 μm to ensure stability of the solution, as demonstrated by a number of studies (*Prassl et al., 2009*; *Plank et al., 2008*).

Fiber orientations, unique to each patient-specific geometry, were applied to each computational mesh on a per-element basis using a rule-based approach (*Bayer et al., 2012*) validated using human fiber orientations acquired in histological and diffusion-tensor MRI studies. This methodology uses the Laplace–Dirichlet method to define apicobasal and transmural directions for every element in the personalized ventricular meshes followed by bidirectional spherical linear interpolation to assign fiber orientations based on a set of rules.

## Electrophysiological properties in the HCM virtual hearts

The personalized 3D virtual hearts of HCM patients incorporated electrical functions from the cellular scale to the whole heart. Electrophysiological remodeling was incorporated in each virtual heart based on the reconstruction of heterogeneously distributed structural remodeling.

At the cellular level, in regions of nonfibrotic myocardium, the human ventricular myocyte model by *ten Tusscher and Panfilov, 2006* was used, with added representation of $I_{NaL}$ (*O'Hara et al., 2011*), as done in our previous studies (*Cartoski et al., 2019*; *Shade et al., 2020a*, *Arevalo et al., 2016*; *Prakosa et al., 2018*; *Shade et al., 2021*). For regions of diffuse fibrosis, we modified the ionic channel kinetics of the ten Tusscher model based on data reported by *Coppini et al., 2013*. In the latter study, prolonged action potential duration and notch elevation following depolarization were observed in experimental recordings from myocytes in hyperthrophied regions acquired via myectomy. As regions of hypertrophy in the HCM heart are also characterized with diffuse fibrotic remodeling, as per histopathological evidence (*Galati et al., 2016*), in the absence of experimental reports of specific ionic changes in regions of diffuse fibrosis, we used those reported by *Coppini et al., 2013*. Specific changes included 107% increase of $I_{NaL}$ maximal conductance, 19% increase of $I_{CaL}$ maximal conductance, 34% decrease of $I_{Kr}$ maximal conductance, 27% decrease of $I_{Ks}$ maximal conductance, 85% decrease of $I_{to}$ maximal conductance, 15% decrease of $I_{K1}$ maximal conductance, 34% increase of sodium-calcium exchanger activity, and 43% reduction of sarcoplasmic/endoplasmic reticulum calcium ATPase activity. The net results of the changes to the cell model include increased action potential duration at 90% repolarization from 280 to 330 ms (+18%) and diminution of the notch after depolarization. *Figure 1A*, middle, shows the action potentials implemented in regions of fibrotic and nonfibrotic myocardium.

At the tissue level, conductivity values along the longitudinal and transverse fiber directions in fibrotic and nonfibrotic myocardium were the same as previously implemented for nonischemic patient heart models (*Shade et al., 2020b*). Dense fibrosis was considered electrically inexcitable. Once completed, the patient-specific HCM electrophysiological heart models were used to assess the patient's risk of arrhythmia.

## Assessing VA risk in the personalized HCM computational models

Full details regarding the simulation of electrical activity in the virtual hearts can be found in previous publications (*Plank et al., 2008*; *Prakosa et al., 2018*; *Vigmond et al., 2008*). Briefly, these were finite element heart models, where the simulation of electrical activity was performed in a monodomain representation of the myocardium using the software package CARP (https://carp.medunigraz.at/). Each virtual heart was paced sequentially from seven uniformly distributed endocardial LV locations using a validated rapid pacing protocol described in detail in previous studies (*Prakosa et al., 2018*; *Arevalo et al., 2016*; *Cartoski et al., 2019*). Similar to our work on VA risk stratifications for patients with ischemic cardiomyopathy (*Arevalo et al., 2016*), simulation results were analyzed to determine whether reentrant VA was induced in the LV HCM models following rapid pacing from any of the sites. If VA was induced from at least one pacing site in a given personalized HCM virtual heart, the patient was then considered at risk of VA. Simulation results were blind to clinical outcome.

The capability of our HCM virtual-heart technology to predict VA risk was compared to risk scores for prophylactic ICD implantation developed by ACCF/AHA (*Gersh et al., 2011*) and ESC (*O'Mahony et al., 2014*) using the patient clinical data.

## Acknowledgements

The authors are grateful to Dr. Iacoppo Olivotto from Careggi University Hospital, Florence, Italy, for the inspiration and discussions.

## Additional information

### Funding

| Funder | Grant reference number | Author |
|---|---|---|
| National Institutes of Health | R01HL142496 | Natalia A Trayanova |
| National Institutes of Health | R01HL126802 | Natalia A Trayanova |
| National Institutes of Health | HL125239 | Edem Binka |
| National Science Foundation | DGE-1746891 | Ryan P O'Hara |
| Fondation Leducq | | Natalia A Trayanova |

The funders had no role in study design, data collection, and interpretation, or the decision to submit the work for publication.

### Author contributions

Ryan P O'Hara, Conceptualization, Data curation, Formal analysis, Investigation, Methodology, Software, Validation, Visualization, Writing - original draft, Writing - review and editing; Edem Binka, Conceptualization, Data curation, Investigation, Methodology, Project administration, Resources, Supervision; Adityo Prakosa, Conceptualization, Data curation, Investigation, Methodology, Software, Supervision; Stefan L Zimmerman, M Roselle Abraham, Dai-Yin Lu, Data curation, Resources; Mark J Cartoski, Conceptualization, Data curation, Investigation, Methodology, Resources; Patrick M Boyle, Conceptualization, Investigation, Methodology, Project administration, Software; Natalia A Trayanova, Conceptualization, Formal analysis, Funding acquisition, Project administration, Resources, Supervision, Writing - original draft, Writing - review and editing

### Author ORCIDs

Ryan P O'Hara http://orcid.org/0000-0002-7636-1799
Patrick M Boyle http://orcid.org/0000-0001-9048-1239
Natalia A Trayanova http://orcid.org/0000-0002-8661-063X

### Ethics

This retrospective study was approved by the Institutional Review Board (IRB) at Johns Hopkins University. All participants provided written informed consent.

### Decision letter and Author response

Decision letter https://doi.org/10.7554/eLife.73325.sa1
Author response https://doi.org/10.7554/eLife.73325.sa2

## Additional files

### Supplementary files

• Transparent reporting form

### Data availability

Where possible (Figures 2, 4, 5), the raw numerical data used to generate the figures have been provided. The segmented MR images of the left ventricles used to generate the finite-element meshes for the computational models used in this study are available on Dryad (https://doi.org/10.5061/dryad.bk3j9kdd0). Patient-derived data, including the MR images, related to this article are not publicly available to respect patient privacy. Interested parties wishing to obtain these data for non-commercial reuse should contact the corresponding author via email. Upon all reasonable requests for access to these data, the corresponding author will work to pursue negotiation of a Data Transfer and Use Agreement with the requesting party, administrators at the requesting party's institution, Johns Hopkins University and Hospital, and the relevant Institutional Review

Boards. Documentation and instructions on the use of the openCARP cardiac electrophysiology simulator and meshalyzer visualization software (both available via https://opencarp.org/) can be used to precisely reproduce the computational protocol applied to patient-specific left ventricular models in this study.

The following dataset was generated:

| Author(s) | Year | Dataset title | Dataset URL | Database and Identifier |
|---|---|---|---|---|
| O'Hara R | 2021 | Left ventricular MRI segmentations | https://doi.org/10.5061/dryad.bk3j9kdd0 | Dryad Digital Repository, 10.5061/dryad.bk3j9kdd0 |

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
