## [Decision Letter]

**Decision letter after peer review:**

Thank you for submitting your article "Personalized Computational Heart Models with T1-Mapped Fibrotic Remodeling Predict Risk of Sudden Death Risk in Patients with Hypertrophic Cardiomyopathy" for consideration by *eLife*. Your article has been reviewed by 3 peer reviewers, and the evaluation has been overseen by a Reviewing Editor and Balram Bhargava as the Senior Editor. The following individual involved in review of your submission has agreed to reveal their identity: Colleen E Clancy (Reviewer #3).

Essential revisions:

1) Please address all of the methodological clarifications addressed by Reviewer 1.

2) Please address all of the conceptual issues raised by Reviewer 2 in the Results/Discussion where appropriate, and more specifically in the Limitations section. Please address the first point in the Recommendations for the Authors section if possible.

*Reviewer #2 (Recommendations for the authors):*

This is a well written manuscript that describes novel work. However, there are some underlying concerns that should be addressed in a revised manuscript.

1. There is no evidence in the manuscript that the simulated electrical activation and recovery in the models bears any relation to electrical activation and recovery in the patients' hearts. Of course this evidence is difficult to obtain in detail unless the patients undergo electrophysiology study. However, if it were possible to compare some features of a simulated ECG signal with real recordings from the patient group, and demonstrate that the models show features consistent with HCM (changes to Q wave, ST segment and T wave) then this would give confidence in the utility of the model.

2. It is possible that the electrophysiology model is not needed to predict arrhythmia risk, and that it is the distribution of diffuse and focal fibrosis that is important. The authors have addressed this question to some extent in Figure 2, but I suspect that it is the shape and position of diffuse fibrosis that is important, not the total amount present.

3. If this is the case, then it may be that a machine learning approach based on imaging alone may be as good as, and quicker than, the model for risk assessment. The authors should consider this possibility.

4. Although the patient data used in this study will be made available on reasonable request, I would encourage wider availability of the patient specific meshes as well as analysis and simulation codes in line with other groups.

Comments on writing and presentation:

The abstract and introduction are both written using very technical language that may not be appropriate for the more general readership of *eLife*. Please consider including a more accessible paragraph that sets the scene for the non-expert.

The data shown in Figure 4B are not suitable for linear regression because the number of unique VA morphologies is a categorical not a continuous variable. To make an inference from these data (which may be highly questionable in any case), please choose a more suitable method.

*Reviewer #3 (Recommendations for the authors):*

The methodology used here is sound and has been well tested by this group. One aspect of this study that should perhaps be highlighted in the Discussion is whether in the absence of a simulated approach as used here, would it be possible to unequivocally link diffuse fibrosis with ventricular arrhythmias based on computational approaches alone?

---

## [Author Response]

Reviewer #2 (Recommendations for the authors):This is a well written manuscript that describes novel work. However, there are some underlying concerns that should be addressed in a revised manuscript.1. There is no evidence in the manuscript that the simulated electrical activation and recovery in the models bears any relation to electrical activation and recovery in the patients' hearts. Of course this evidence is difficult to obtain in detail unless the patients undergo electrophysiology study. However, if it were possible to compare some features of a simulated ECG signal with real recordings from the patient group, and demonstrate that the models show features consistent with HCM (changes to Q wave, ST segment and T wave) then this would give confidence in the utility of the model.

Our virtual heart models were used purely for the assessment of arrhythmia inducibility in the substrate, and we did not aim to replicate VT in the inducible patients. The rapid pacing protocol we used is not comparable to sinus rhythm, so we cannot make such comparisons. The utility of our model rests on its ability to predict *likelihood of arrhythmia* in the substrate and not on it replicating the patient’s rhythm features or specific VTs.

2. It is possible that the electrophysiology model is not needed to predict arrhythmia risk, and that it is the distribution of diffuse and focal fibrosis that is important. The authors have addressed this question to some extent in Figure 2, but I suspect that it is the shape and position of diffuse fibrosis that is important, not the total amount present.

We could not agree more with the reviewer. We have shown here that the arrhythmogenic propensity of the ventricular substrate with diffuse fibrosis is patient-specific; it is indeed the amount of fibrosis and its *distribution* (shape and location) that determine inducibility of arrhythmia. We have further emphasized this in the Discussion (lines 333-5).

3. If this is the case, then it may be that a machine learning approach based on imaging alone may be as good as, and quicker than, the model for risk assessment. The authors should consider this possibility.

Absolutely agree! Since we have only 26 patients, it is not possible to discern specific features of the fibrotic distribution that are most likely to predict arrhythmogenesis. In the Discussion of the original manuscript, we mentioned that this could be achieved with machine learning in a large cohort (the same text can be found on lines 365-9 of the revised paper). We are in the process of collecting data on a large HCM cohort for a machine learning study to address this question. However, as we expect to determine what features of fibrosis distribution are most important in HCM patients, it is important to keep in mind the mechanistic insight provided by the simulation study as it will provide explainability of the features learned by the algorithm.

4. Although the patient data used in this study will be made available on reasonable request, I would encourage wider availability of the patient specific meshes as well as analysis and simulation codes in line with other groups.

Thank you for the suggestion -- we have already done that. After the paper was sent for review, the journal asked us to provide the meshes and other analysis. Data has been provided via Dryad at request of the journal and the link will be made publicly active upon paper acceptance.

Comments on writing and presentation:The abstract and introduction are both written using very technical language that may not be appropriate for the more general readership of eLife. Please consider including a more accessible paragraph that sets the scene for the non-expert.

We have made some adjustments to the text (lines 21-2) to make it more accessible to the non-expert. Thank you for the suggestion.

The data shown in Figure 4B are not suitable for linear regression because the number of unique VA morphologies is a categorical not a continuous variable. To make an inference from these data (which may be highly questionable in any case), please choose a more suitable method.

Thank you for the suggestion. We have replaced the correlation analysis with a logistic regression in figure 4B and updated the text in lines 203 and 220.

Reviewer #3 (Recommendations for the authors):The methodology used here is sound and has been well tested by this group. One aspect of this study that should perhaps be highlighted in the Discussion is whether in the absence of a simulated approach as used here, would it be possible to unequivocally link diffuse fibrosis with ventricular arrhythmias based on computational approaches alone?

Thank you for this comment. We wouldn’t argue that one can link unequivocally diffuse fibrosis with ventricular arrhythmias based on computational approaches alone. It is the actual fibrotic distribution that determines the arrhythmogenic propensity. A machine learning approach using the MRI images may allow us to determine which features of diffuse fibrosis are most important for arrhythmogenesis. We have made comments along these lines in the Discussion (lines 365-9).